# Trends, and patterns, of premarital sexual intercourse and its associated factors among never-married young women aged 15–24 in Sierra Leone

**Augustus Osborne**[1]*, **Castro Ayebeng**[2,3], **Peter Bai James**[4,5], **Camilla Bangura**[1], **Richard Gyan Aboagye**[6], **Bright Opoku Ahinkorah**[7,8,9]

1 Department of Biological Sciences, School of Environmental Sciences, Njala University, PMB, Freetown, Sierra Leone, 2 Department of Population and Health, College of Humanities and Legal Studies, University of Cape Coast, Cape Coast, Ghana, 3 Department of Research and Advocacy, Challenging Heights, Winneba, Ghana, 4 Faculty of Health, National Centre for Naturopathic Medicine, Southern Cross University, Lismore, Australia, 5 Faculty of Pharmaceutical Sciences, College of Medicine and Allied Health Sciences, University of Sierra Leone, Freetown, Sierra Leone, 6 Department of Family and Community Health, Fred N. Binka School of Public Health, University of Health, and Allied Sciences, Hohoe, Ghana, 7 REMS Consultancy Services, Takoradi, Sekondi-Takoradi, Ghana, 8 School of Clinical Medicine, University of New South Wales Sydney, Sydney, Australia, 9 Faculty of Health, School of Public Health, University of Technology Sydney, Sydney, Australia

* augustusosborne2@gmail.com

## Abstract

### Background

Premarital sexual intercourse has essential implications for the sexual and reproductive health and rights of young women. These include increased sexual pleasure and satisfaction as well as exposure to the risks of unintended pregnancy and sexually transmitted infections, including HIV/AIDS. This study examined the trends, patterns, and associations of premarital sexual intercourse among young women aged 15–24 in Sierra Leone.

### Methods

Nationally representative cross-sectional data from the 2008, 2013, and 2019 Demographic and Health Surveys in Sierra Leone were used for the study. A weighted sample of 9,675 never-married young women was used to estimate the pooled prevalence of premarital sexual intercourse in Sierra Leone. Percentages were used to present the results of the trends and patterns of premarital sexual intercourse. We employed a multilevel binary logistic regression modelling technique to examine the associations of premarital sexual intercourse. The results were presented using adjusted odds ratio with their respective 95% confidence interval.

### Results

The pooled prevalence of premarital sexual intercourse among the young women in Sierra Leone was 62.9%. Over the survey years, premarital sexual intercourse increased from

**Data Availability Statement:** Third party data was obtained for this study from The DHS Program (https://dhsprogram.com/). Data may be requested

from The DHS Program after creating an account and submitting a concept note. More access information can be found on The DHS Program website (https://dhsprogram.com/data/Access-Instructions.cfm).The data set is openly available upon permission from the MEASURE DHS website (https://www.dhsprogram.com/data/available-datasets.cfm). The authors confirm that interested researchers would be able to access these data in the same manner as the authors. The authors also confirm that they had no special access privileges that others would not have.

**Funding:** The author(s) received no specific funding for this work.

**Competing interests:** The authors have declared that no competing interests exist.

**Abbreviations:** AIC, Akaike Information Criterion; aOR, Adjusted Odds Ratio; CI, Confidence Intervals; COR, Crude Odds Ratio; ICC, Intraclass Correlation; PCV, Proportional Change in Variance; PSI, Premarital Sexual Intercourse; SE, Standard Error.

59.8% in 2008 to 65.1% in 2013. However, it declined by 3.5% to 61.6% in 2019. Young women aged 20–24 (aOR = 12.47, 95% CI = 10.54–14.76) had higher odds of engaging in premarital sexual intercourse than those aged 15–19. Young women with higher educational levels (aOR = 1.87, 95% CI = 1.17–2.99), those who were working (aOR = 1.60, 95% CI = 1.44–1.78), those who listened to the radio (aOR = 1.33, 95% CI = 1.29–1.60), and those who lived in the Northwestern (aOR = 2.19, 95% CI = 1.68–2.84), Eastern (aOR = 1.47, 95% CI = 1.23–1.760, Northern (aOR = 1.48, 95% CI = 1.25 -, 1.76), and Southern (aOR = 1.63, 95% CI = 1.36–1.94) regions were more likely to engage in premarital sexual intercourse compared to those with no formal education, those not working, those who did not listen to the radio, and those who lived in the Western region, respectively. Young women in the richest wealth category (aOR = 0.62, 95% CI = 0.49–0.78), and residing in rural areas (aOR = 0.84, 95% CI = 0.72–0.98) had lower odds of engaging in premarital sexual intercourse relative to those from the poorest wealth quintile and those living in urban areas.

## Conclusion

Our study found a high prevalence of premarital sexual intercourse among young women in Sierra Leone. Premarital sexual intercourse was associated with age, educational level, wealth, employment, and region. This necessitates providing them with comprehensive information regarding sexual and reproductive health behaviours, specifically emphasising the benefits and adverse consequences of engaging in sexual experimentation. Additionally, it is crucial to promote the adoption of abstinence, injections, implants, and condom usage through consistent advocacy for youth-risk communication.

## Introduction

Premarital sexual intercourse (PSI) is defined as engaging in sexual intercourse before marriage or before the intended marriage [1–4]. PSI is a common phenomenon among young people(15-24years) in many parts of the world, especially in sub-Saharan Africa (SSA), where marriage is often delayed due to economic, social, and cultural factors [2]. It can expose young women to various risks that may differ from sex during marriage in terms of consent, partner, pregnancy, abortion, and social acceptance. Therefore, young women should be informed and empowered to make responsible and healthy decisions about their sexual and reproductive health and rights. PSI has essential implications for the sexual and reproductive health and rights of young women. Notable implications of PSI include unintended pregnancy, unsafe abortion, sexually transmitted infections (STIs), including, HIV/AIDS, and gender-based violence [2,5–7]. PSI may affect young women's educational attainment and economic opportunities, as well as their marital prospects and social status [8]. Also, PSI can causeearly pregnancy and motherhood which can lead to school dropout, and limited educational opportunities and future earnings. Additionally, stigma and discrimination associated with PSIcan lead to social isolation and lower social status in certain societies [8].

PSI is prevalent among young women in SSA, with variations across countries and regions, reflecting the diversity of sociocultural and economic contexts [9]. According to a recent study by Budu et al. [9], the prevalence of PSI among young women aged 15–24 in SSA was 39.4%, ranging from 5.0% in Comoros to 75.3% in Liberia. In the same study, the prevalence of PSI in

Sierra Leone was 61.5%. Evidence suggests that several individual, household, and community-level factors associated with PSI includeage, education, wealth, religion, media exposure, place of residence, and sub-regional location [9,10–12].

Additional studies has also identified three distinct categories of predictors for PSI among young women in SSA: individual-level, family-level, and institutional-level predictors [10,11]. The individual-level predictors encompass age, gender, ethnicity, romantic involvement, and feelings of loneliness [10]. The predictors at the family level includefamily type, income, occupation, broken families, and parenting. On the other hand, the predictors at the institutional level v rules and laws that extend beyond a single social network. These predictors also include other forms of communication such as mobile phones, internet, books and magazines, radio, and television [10,12]. Furthermore, previous studies have shown a correlation between several socio-environmental factors and the involvement of young women in risky sexual behaviours, such as transactional sex and engaging in sexual relationships with multiple partners. These factors include but are not limited to unfavourable urban environments, elevated unemployment rates, precarious income stability, limited literacy levels, and insufficient access to recreational amenities [13–15].

Several studies in Sierra Leone have mainly focused on sexual practices and behaviours among adolescents and adults combined [16–19]. However, there is limited evidence on trends, patterns, and associations of PSI among young women in Sierra Leone. Sierra Leone has one of the lowest rates (48.6%) of literacy and one of the highest rates (22.1%) of teenage pregnancy in the world [20]. Understanding the trends, patterns, and associations of PSI among young women in Sierra Leone is crucial for designing and implementing effective interventions to promote their sexual and reproductive health and rights. Therefore, this study aims to fill this knowledge gap by using nationally representative data from the Sierra Leone Demographic and Health Surveys (SLDHS) conducted in 2008, 2013, and 2019 to examine the trends and associations of PSI among young women aged 15–24 in Sierra Leone. By comparing data across multiple periods, we aimed to discern whether observed changes in PSI are consistent or transient, thus providing more robust insights into the dynamics of PSI. Additionally, repeated cross-sectional studies help account for variations in demographic factors, ensuring that findings are representative and generalizable to the broader population.

## Materials and methods

### Study area and data source

Sierra Leone is in West Africa and is administratively divided into five regions: the Western, Northern, Eastern, Southern, and Northwestern. In 2024, Sierra Leone's population stands at 8,977,972, reflecting a 2.13% growth compared to the previous year [21]. Sierra Leone has a Muslim-majority population, constituting 78%, while Christians make up 21%, with the remaining 1% belonging to other religions [22]. Additionally,approximately 42% of its population are under 15 years of age and 19% are young people aged 15–24 years [22]. The primary ethnic groups include the Mende, Temne, Limba, and Krio. We employed a secondary analysis of a pooled dataset from the 2008, 2013, and 2019 SLDHS. The SLDHS are nationally representative cross-sectional surveys with a sample of 7,374, 16,658, and 15,574, respectively. The SLDHS is a component of the global DHS series and aims to provide nationally representative data from developing countries, focusing on women between the ages of 15 and 49 years [22]. A stratified, two-stage cluster sampling design was employed to derive nationwide representativeness. The sampling frame for the survey was based on the 2004 and 2015 Sierra Leone Population and Housing Census. For instance, the 2008 and 2013 surveys and the 2019 surveys were based on the 2004 and 2015 Population and Housing Census, respectively [22–24]. The

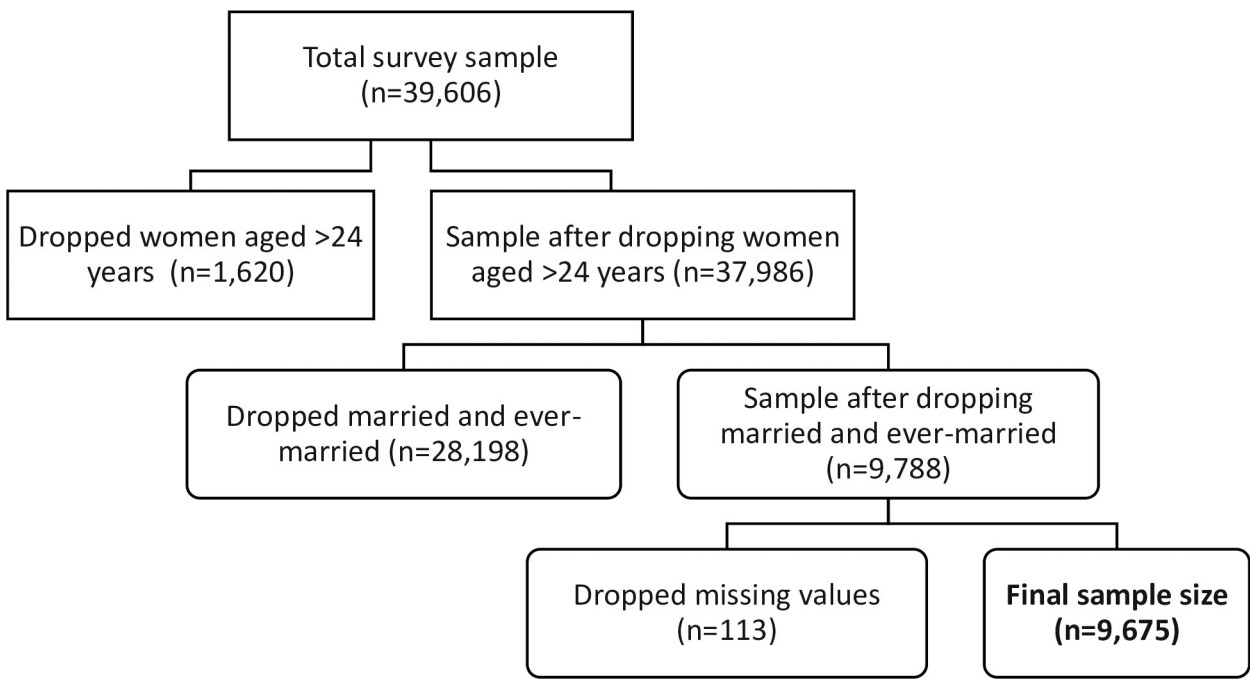

**Fig 1. Flow chart of sample determination.**

first stage involved selecting enumeration areas (EAs) based on probability proportional to size. Later,a systematic sampling method was used to select households from each EA. A detailed description of the sampling procedure can be found in the 2008, 2013, and 2019 SLDHS [22–24]. Standardised structured questionnaires were used to collect data from the respondents on health and social indicators, including age at first sexual intercourse. The final analysis of this study included a total weighted sample of 9,675 never-married young women aged 15–24 years, with specific contributions of 1,178, 4,236, and 4,261 from the 2008, 2013 and 2019 surveys, respectively (see Fig 1).

## Study variables

**Outcome variable.**   The study focused on PSI as the key outcome variable. In this study, PSI was operationalised as any young women engaging in sexual intercourse before marriage. The PSI variable was derived from the question, "At what age did you have your first sex?" The study classified the young who had never had sex as "not engaged in PSI" and coded as "0," while those who had sex at age eight or above were classified as "ever had PSI" and coded as "1."

**Explanatory variable.**   Multiple individual-level and contextual factors informed by empirical literature relating to PSI [9,25,26] were selected for the analyses as the explanatory variables. The variables included in the studywere age, level of education, religion, employment status, wealth status, and exposure to the radio, magazines/newspapers, and television. Age was categorised as 15–19 = 0 and 20–24 = 1. Levels of education were coded as no formal education = 0, primary = 1, secondary = 2, and higher education = 3. Religion was categorised as Christian = 0, Islam = 1, and Other = 2. Employment status was coded as not working = 0 and working = 1. Wealth status was coded as poorest = 0, poorer = 1, middle = 2, richer = 3, and richest = 4. Exposure to the radio, magazines/newspapers, and television was coded as yes = 1 (if exposed less than once a week or at least once a week) and no = 0 (if not exposed at

all). The contextual variables include place of residence, which was coded as urban = 0 and rural = 1. And region coded as Eastern = 0, Northern = 1, Southern = 2, Western = 3, and Northwestern = 4.

**Statistical analysis.** We conducted a detailed analysis of the percentage of young women who had PSI, considering various data points and a pool of the dataset. The results information were presented using a line graph. Additionally, we provided a descriptive breakdown of the proportional distribution of PSI based on background characteristics. A chi-square test was employed to ascertain statistically significant associations between explanatory variables and PSI. Next, we presented the results of the regional patterns of PSI in percentages, using a bar graph. Due to the hierarchical structure of the SLDHS data, with respondents nested within clusters, a more sophisticated analysis is appropriate to utilise a multilevel logistic regression model rather than the traditional logistic regression model [27]. Four distinct multilevel logistic regression models were constructed: the null model (0), which had no explanatory variable, was fitted to show the variance in PSI attributed to the primary sampling units (PSUs); Model I, included individual-level factors (age, education, religion, wealth index, employment status, exposure to radio, magazines/newspapers, and radio); Model II, captured contextual variables (residence and region); and the final model adjusted for both individual and contextual -level variables. The regression results were presented using adjusted odds ratios(aOR) with their respective 95% confidence intervals (CI). To evaluate the fitness of each model, we employed the log-likelihood ratio (LLR) test, Akaike's Information Criterion (AIC), and Bayesian Information Criterion (BIC). The model characterised by the lowest AIC was chosen as the best fit in the analysis. Before fitting the multilevel logistic regression model, we assessed the potential for multicollinearity using the variance inflation factor (VIF). The mean VIF score of 4.87 indicated the absence of significant multicollinearity. The women's sample weight provided in the SLDHS dataset was used to generate the estimates. Application of the sample weight is essential when generating estimates because it adjusts for non-response and over- and under sampling of respondents during the survey. All statistical analyses were conducted using Stata version 17.

**Fixed and random effect estimates.** The analysis incorporated both fixed and random effects. In the fixed effect analysis, the association between the explanatory variables and PSI. Conversely, the random effect analysis assessed the variations between clusters (EAs). We calculated two essential metrics to quantify these variations: the Intra-class correlation coefficient (ICC) and the proportional change in variance (PCV). The ICC gauges the extent of variation within clusters, specifically among individuals within the same cluster. It was computed using the formula:

ICC = $[V_A/ (V_A + \pi2/3)] = V_A/ (V_A + 3.29)$. Here, $V_A$ represents the estimated variance in each model [27].

To assess the overall variation attributed to individual and contextual factors in each model, we utilised the proportional change in variance (PCV), calculated as:

PCV = $(V_A−V_B)/V_A$. In this equation, $V_A$ is the variance of the initial model, and $V_B$ is the variance of the model with additional terms [27].

## Ethics approval and informed consent

No ethical clearance was sought for the current study. However, ethical standards are ensured during DHS surveys. The SLDHS 2019 survey protocol was reviewed and approved by the Sierra Leone Ethics and Scientific Review Committee and the ICF Institutional Review Board. Written informed consent was obtained from the respondents, and written informed consent was obtained from legally authorised representatives of minor respondents.

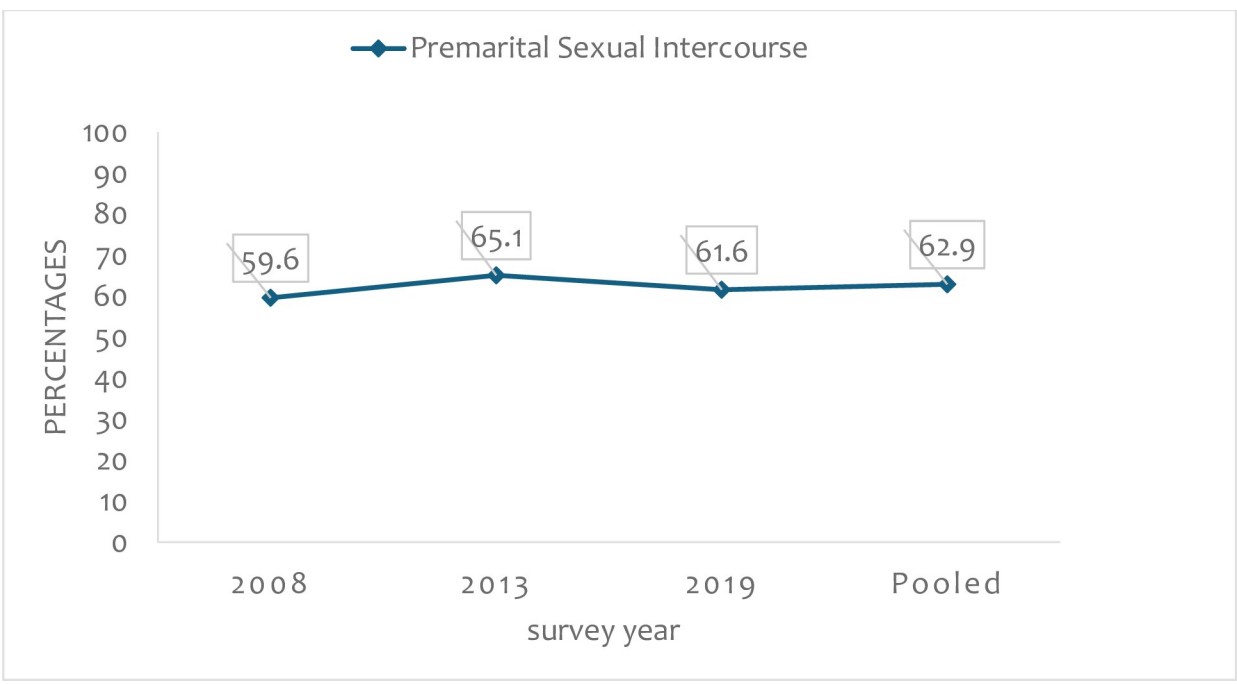

**Fig 2. A graph showing percentages for trends of PSIamong young women in Sierra Leone 2008–2019 and pooled dataset.**

## Results

### Trends of PSI among young women in Sierra Leone

Fig 2 displays the trends of PSI among young women in Sierra Leone. The pooled prevalence of PSI among young women in Sierra Leone was 62.9%. The survey year-specific prevalence indicated 59.8% in 2008, which increased to 65.1% in 2013. However, the prevalence of PSI declined by 3.5% to 61.6% in 2019.

### Distribution of PSIacross the explanatory variables per survey years

The results in Table 1 show the distribution of PSI across the different survey years (2008, 2013, and 2019). The percentage of young women who engaged in PSI was higher among the older age group 20–24 (84.1%, 93.3%, and 92.2%) than the younger age group 15–19 in all survey years. The percentage of young women who engaged in PSI was higher among the higher-educated women (95.5%, 87.8%, and 84.7%) than the lower-educated women in all survey years. Regarding religion, the percentage of young women who engaged in PSI was higher among Christian young women(64.0%, 70.8%, and 65.3) than Muslim women or women of other religions in all survey years. The percentage of young women who engaged in PSI was higher among employed women(60.3%, 69.7%, and 73.6) than the unemployed women in all survey years. Moreover, the percentage of young women who engaged in PSI was higher among the wealthier young women(66.2%, and 69.7%) than the poorer young women in 2008 and 2013 but lower (63.3%) in 2019. Concerning media exposure, the percentage of young women who engaged in PSI was higher among those exposed to newspapers/magazines (64.2%, 71.1% and 64.4%), radio(60.8%, 67.2% and 67.1%), or television(62.1%, 66.7% and 61.2%) than those who were not exposed in all survey years. The percentage of young women who engaged in PSI was higher among urban women(63.1%, and 66.4%) than rural women in 2008 and 2013 but lower (60.3%) in 2019. Age, educational level, religion, employment status,

**Table 1. Distribution of PSI among young women aged 15–24 across the background characteristics per survey years(2008–2019) in Sierra Leone.**

| Explanatory variables | 2008 | 2013 | 2019 | Pooled dataset |
|---|---|---|---|---|
| | n (%) | n (%) | n (%) | n (%) |
| **Age** | P<0.001 | P<0.001 | P<0.001 | P<0.001 |
| 15–19 | 444 (49.6) | 1,812 (55.9) | 1,432 (49.0) | 3,683 (52.2) |
| 20–24 | 307 (84.1) | 987 (93.3) | 1,102 (92.2) | 2,400 (91.6) |
| **Educational level** | P<0.001 | P<0.001 | P<0.001 | P<0.001 |
| No education | 120 (52.8) | 410 (68.4) | 246 (61.2) | 770 (63.2) |
| Primary | 151 (51.2) | 320 (43.5) | 282 (45.9) | 749 (45.8) |
| Secondary | 441 (63.3) | 1,948 (68.9) | 1,899 (63.8) | 4,297 (65.9) |
| Higher | 39 (95.5) | 121 (87.8) | 107 (84.7) | 266 (87.4) |
| **Religion** | P = 0.014 | P = 0.001 | P<0.001 | P<0.001 |
| Christian | 268 (64.0) | 826 (70.8) | 712 (65.3) | 1,802 (67.5) |
| Islam | 480 (57.6) | 1,959 (63.0) | 1,820 (60.2) | 4,263 (61.1) |
| Other | 3 (40.8) | 14 (60.7) | 1(29.6) | 17 (54.0) |
| **Employment status** | P = 0.767 | P<0.001 | P<0.001 | P<0.001 |
| Unemployed | 468 (59.2) | 1,628 (62.2) | 1,452 (54.9) | 3,545 (58.5) |
| Employed | 283 (60.3) | 1,171 (69.7) | 1,081 (73.6) | 2,538 (70.2) |
| **Wealth index** | P = 0.001 | P = 0.009 | P = 0.003 | P<0.001 |
| Poorest | 51 (43.9) | 343 (62.0) | 270 (64.9) | 665 (61.3) |
| Poorer | 72 (56.0) | 320 (60.9) | 319 (63.5) | 713 (61.5) |
| Middle | 93 (57.4) | 443 (68.3) | 473 (64.5) | 1,014 (65.4) |
| Richer | 185 (66.2) | 710 (69.7) | 747 (63.3) | 1,646 (66.2) |
| Richest | 348 (61.0) | 983 (63.3) | 724 (56.4) | 2,044 (60.2) |
| **Exposed to reading newspaper** | P = 0.008 | P<0.001 | P = 0.004 | P<0.001 |
| No | 487 (57.4) | 2,155 (63.2) | 2,210 (61.1) | 4,827 (61.6) |
| Yes | 264 (64.2) | 684 (71.7) | 323 (64.4) | 1,256 (68.1) |
| **Exposed to listening to the radio** | P = 0.094 | P<0.001 | P<0.001 | P<0.001 |
| No | 216 (56.9) | 713 (59.7) | 1,259 (56.8) | 2,208 (57.7) |
| Yes | 535 (60.8) | 2,086 (67.2) | 1,274 (67.1) | 3,875 (66.3) |
| **Exposed to watching television** | P = 0.247 | P = 0.005 | P = 0.169 | P = 0.002 |
| No | 476 (58.3) | 1,837 (64.3) | 1,624 (61.7) | 3,937 (62.5) |
| Yes | 274 (62.1) | 963 (66.7) | 909 (61.2) | 2,146 (63.6) |
| **Place of residence** | P = 0.017 | P = 0.037 | P = 0.841 | P = 0.061 |
| Urban | 500 (63.1) | 1,463 (66.4) | 1,508 (60.3) | 3,471 (63.1) |
| Rural | 251 (53.7) | 1,336 (63.8) | 1,025 (63.5) | 2,612 (62.6) |

*Note*: Estimates are weighted; P = (P-value from Chi-square test).

wealth index, newspaper, radio, and television exposure were all associated with PSI in the pooled dataset.

## Patterns of PSI among young women across different survey years by region

Fig 3 shows the proportion of PSI among young women in Sierra Leone across different survey years by region. PSI among young women in Sierra Leone was relatively high across all regions per survey years. In 2008, the highest prevalence of PSI was reported among young women in the Eastern region (62.7%), with the least prevalence among those in the Southern region (58.2%). The proportion of PSI was highest among young women residing in the Northern

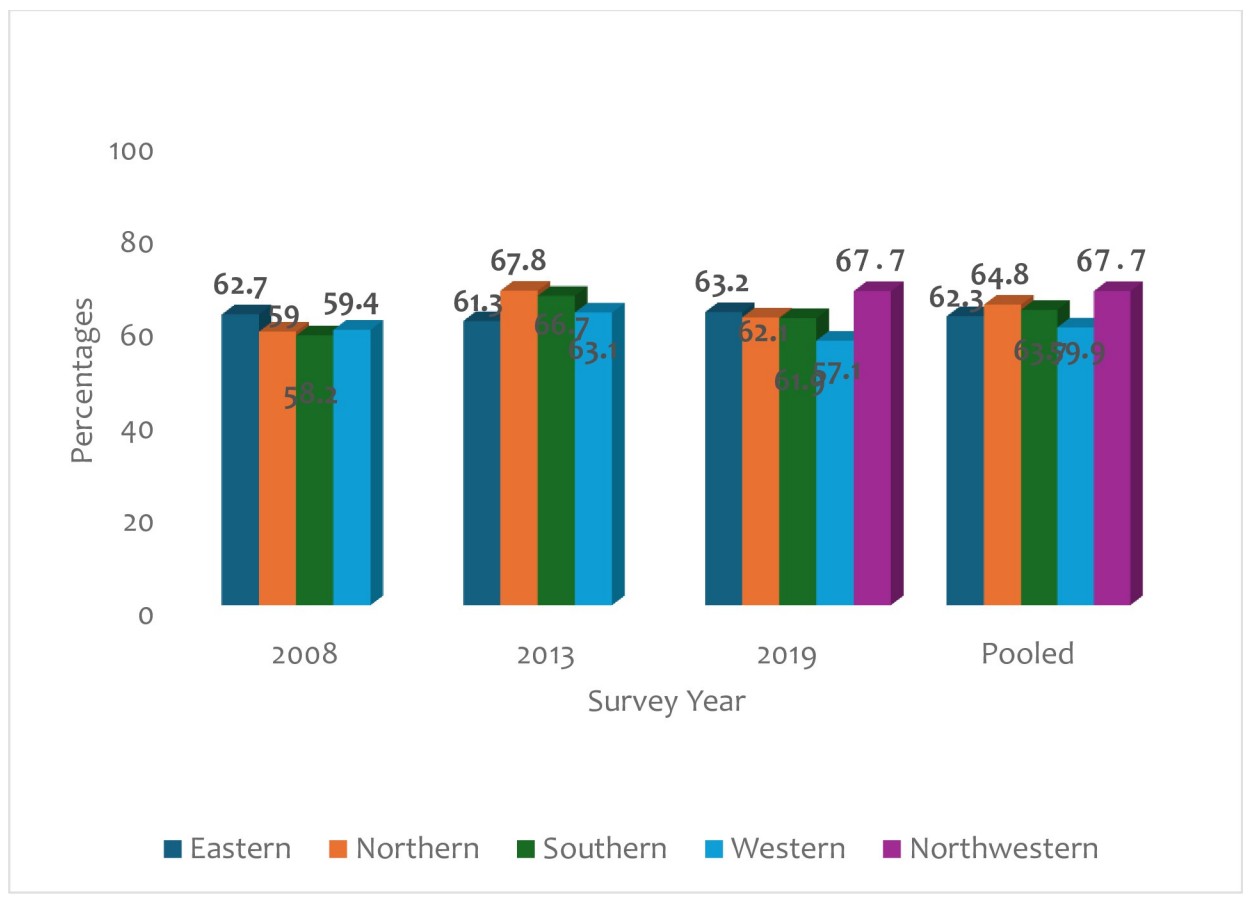

**Fig 3. A graph showing the percentages of PSI among young women aged 15–24 across different survey years(2008–2019) by region in Sierra Leone.**

region (67.8%) in 2013. The Northwestern region (67.7%) recorded the highest prevalence of PSI among young women in 2019. At the pooled level, the region with the highest prevalence of PSI was Northwestern (67.7%), with the lowest proportion among those from the Western region (59.9%).

## Factors associated withPSI among young women in Sierra Leone

Table 2 presents the results of the factors associated withPSI among young women in Sierra Leone. Young women aged 20–24 (aOR = 12.47, 95% CI = 10.54–14.76) had higher odds of engaging in PSI than those aged 15–19. Compared to young women with no education, the odds of PSI was lower among those with primary education (aOR = 0.62, 95% CI = 0.52–0.73). However, those in secondary education (aOR = 1.46, 95% CI = 1.24–1.71) and those with higher educational levels (aOR = 1.87, 95% CI = 1.17–2.99) were more likely to engage in PSI. Working young women (aOR = 1.60, 95% CI = 1.44–1.78) were more likely to engage in PSI than those not working. Young women in the richest wealth category (aOR = 0.62, 95% CI = 0.49–0.78) had lower odds of engaging in PSI relative to those in the poorest category. Young women who listened to the radio (aOR = 1.33, 95% CI = 1.29–1.60) were more likely to engage in PSI than those who did not listen. Young women residing in rural areas (aOR = 0.84, 95% CI = 0.72–0.98) had lower odds of engaging in PSI compared to those in urban areas. At the regional level, young women residing in the Eastern (aOR = 1.47, 95%

**Table 2. Multilevel logistic regression analysis of factors associated with PSI among young women aged 15–24 in Sierra Leone.** (pooled dataset).

| Variables | Null model (0) | Model I | Model II | Model III |
|---|---|---|---|---|
| | | aOR (95% CI) | aOR (95% CI) | aOR (95% CI) |
| **Age** | | | | |
| 15–19 | | 1.00 | | 1.00 |
| 20–24 | | 11.92*** [10.09–14.09] | | 12.47*** [10.54–14.76] |
| **Educational level** | | | | |
| No education | | 1.00 | | 1.00 |
| Primary | | 0.61*** [0.52–0.73] | | 0.62*** [0.52–0.73] |
| Secondary | | 1.45*** [1.24–1.69] | | 1.46*** [1.24–1.71] |
| Higher | | 1.76* [1.10–2.82] | | 1.87** [1.17–2.99] |
| **Religion** | | | | |
| Christian | | 1.41 [0.56–3.54] | | 1.57 [0.62–3.96] |
| Islam | | 1.17 [0.47–2.93] | | 1.28 [0.51–3.21] |
| Other | | 1.00 | | 1.00 |
| **Employment status** | | | | |
| Unemployed | | 1.00 | | 1.00 |
| Employed | | 1.60*** [1.44–1.78] | | 1.60*** [1.44–1.78] |
| **Wealth index** | | | | |
| Poorest | | 1.00 | | 1.00 |
| Poorer | | 1.06 [0.88–1.29] | | 1.08 [0.89–1.31] |
| Middle | | 1.18 [0.98–1.41] | | 1.18 [0.99–1.42] |
| Richer | | 0.97 [0.82–1.16] | | 0.95 [0.78–1.15] |
| Richest | | 0.59*** [0.49–0.71] | | 0.62*** [0.49–0.78] |
| **Exposed to reading newspaper** | | | | |
| No | | 1.00 | | 1.00 |
| Yes | | 1.11 [0.97–1.27] | | 1.12 [0.98–1.29] |
| **Exposed to listening to the radio** | | | | |
| No | | 1.00 | | 1.00 |
| Yes | | 1.44*** [1.29–1.60] | | 1.33*** [1.19–1.49] |
| **Exposed to watching television** | | | | |
| No | | 1.00 | | 1.00 |
| Yes | | 0.93 [0.83–1.06] | | 1.00 [0.89–1.14] |
| *Community-level factors* | | | | |
| **Place of residence** | | | | |
| Urban | | | 1.00 | 1.00 |
| Rural | | | 0.82*** [0.74–0.91] | 0.84* [0.72–0.98] |
| **Region** | | | | |
| Eastern | | | 1.23** [1.06–1.43] | 1.47*** [1.23–1.76] |
| Northern | | | 1.27** [1.10–1.46] | 1.48*** [1.25–1.76] |
| Southern | | | 1.28** [1.10–0.48] | 1.63*** [1.36–1.94] |
| Western | | | 1.00 | 1.00 |
| Northwestern | | | 1.52*** [1.21–1.89] | 2.19*** [1.68–2.84] |
| **Survey year** | | | | |
| 2008 | | | | 1.00 |
| 2013 | | | | 1.28** [1.10–1.50] |
| 2019 | | | | 0.94 [0.80–1.11] |
| **Random effects results** | | | | |
| Variance (SE) | 0.114 (0.024) | 0.160 (0.031) | 0.113 (0.023) | 0.140 (0.029) |

(*Continued*)

**Table 2.** (Continued)

| Variables | Null model (0) | Model I | Model II | Model III |
|---|---|---|---|---|
| | | aOR (95% CI) | aOR (95% CI) | aOR (95% CI) |
| ICC (%) | 3.34 | 4.63 | 3.33 | 4.09 |
| PCV (%) | 1.00 | -45.35 | 29.37 | 64.60 |
| **Model fit statistics** | | | | |
| Log-likelihood | -6345.80 | -5354.94 | -6334.10 | -5318.91 |
| AIC | 12695.60 | 10741.88 | 12682.20 | 10683.82 |
| BIC | 12709.96 | 10856.72 | 12732.45 | 10848.90 |

aOR: Adjusted odds ratio; 95% CI: 95% Confidence Interval; 1.00: Reference category, SE: Standard error, ICC: Intraclass correlation, PCV: Proportional change in variance; AIC: Akaike Information Criterion; Bayesian Information Criterion.

CI = 1.23–1.76), Northern (aOR = 1.48, 95% CI = 1.25–1.76), Southern (aOR = 1.63, 95% CI = 1.36–1.94), and Northwestern (aOR = 2.19, 95% CI = 1.68–2.84) regions were more likely to engage in PSI compared to those in the Western region. Women in the 2013 survey year (aOR = 1.28, 95% CI = 1.10–1.50) were more likely to engage in PSI than those in the 2008 survey year.

## Random effect results

Approximately 33% of the prevalence of PSI was attributed to the variations between the clusters (ICC = 3.34). The between-cluster difference increased to 46.3% in Model I, decreased to 33.3% in Model II, and increased to 40.9% in Model III. These ICC results suggest that the variations in the likelihood of PSI can be attributed to the variances across the clusters. The AIC values showed a similar U-shaped pattern as the ICC values, reaching their lowest point in model III. Therefore, model III was chosen as the most suitable model for analysing the factors associated with PSI among young women in Sierra Leone.

## Discussion

The present study investigated the trends and patterns of PSI and its associated factors among young women in Sierra Leone. The trends and patterns of PSI in Sierra Leone increased from 2008 to 2013 but decreased in 2019. The study also revealed that the overall prevalence of PSI among young women in Sierra Leone was 62.9%. Our finding is similar to the results of a previous study in which high PSI prevalence was reported in Angola (64.0%), Congo (64.8%), and Namibia (62.7%) [9]. Possible reasons for the high prevalence of PSIamong young women 15–24 years in Sierra Leone include cultural norms, lack of education, economic factors, and social pressures. Addressing these issues through comprehensive sex education, empowerment, and supportive policies is crucial for promoting healthier outcomes for young women [28,29]. Poverty may be another major driver of PSIamong young women in Sierra Leone, as they may engage in transactional sex or exchange sex for money, food, or other material benefits. Poverty also limits their access to education, health care, and family planning services, which can reduce their ability to make informed decisions about their sexuality and reproductive health [9,30]. Education is another key factor that can empower young women to delay their sexual debut, avoid unwanted pregnancies, and pursue their aspirations. However, many young women in Sierra Leone face barriers to education, such as high cost, low quality, gender discrimination, and social norms that favour boys over girls [30]. As a result, many young women drop out of school or never enroll, which increases their vulnerability to PSIand its

consequences [30]. The results of our study suggest that there may be an elevated susceptibility to STIs among young females residing in Sierra Leone due to the high PSI found in the study. Hence, public and private entities must enhance their endeavours in educating young women about the ramifications associated with involvement in PSI as education about sexual health and reproductive rights can empower young women to make informed decisions about their bodies and lives.

Consistent with prior research [9,31–34], our study found that age was associated with PSI. The younger age group (15–19 years) might bebe under parental guidance or supervision as they are still not adults, limiting their engagement in sexual intercourse [17]. In Sierra Leone, adulthood is likely at the age of 21, i.e. living with their parents who monitor their daily activities, compared to 20–24 years who are considered adults and have the liberty to make sexual choices such as having sex before marriage [17]. Cultural norms may also influence PSIas it is believed that having children is a sign of fertility and social status, which may pressure some young women aged 20–24 to conceive before marriage [35]. Given the societal expectation for younger women aged 15–19 to be enrolled in educational institutions, it is unsurprising that they are less likely to experience PSI.

The present study revealed a positive association between secondary or higher educational attainment among young women and their likelihood of engaging in PSI, in contrast to those with noformal education, consistent with previous studies [9,31]. Young women with secondary or higher education may have more access to sexual information and media, which could influence their attitudes and behaviours towards sex [36]. Young women with secondary or higher education may have more knowledge and access to contraceptives and family planning services, which could reduce their fear of getting pregnant or contracting STIs [37].

Young women who were working had higher odds of engaging in PSI compared to those who were not working. Working young women may have higher levels of education and empowerment and more financial resources and autonomy to make their own decisions about their sexual and reproductive health and rights [28]. Young women with the highest wealth index showed a decreased likelihood of engaging in PSI compared to those from the poorest wealth index, in contrast to the previous study [38]. Young women with the highest wealth index may face less economic hardship and social vulnerability, which could reduce their need or pressure to engage in PSI [9].

Our study revealed that young women exposed to the media (radio and newspapers) had higher odds of engaging in PSI than those not exposed, similar to the findings of an earlier study [39]. Media exposure may influence young women's attitudes and norms regarding sexuality, gender roles, and relationships. Media may also provide information and education on sexual and reproductive health or challenge harmful practices and stereotypes [17]. Exposure to media may also increase the opportunities and motivations for young women to engage in PSI. For instance, media may expose them to new lifestyles, aspirations, and peer groups or create a sense of curiosity, experimentation, or rebellion [17].

The findings revealed that young women residing in rural areas were less likely to engage in PSI than their counterparts residing in urban areas, contrasting with the findings reported in another research [31,40]. Rural areas tend to be more conservative and traditional in their values and beliefs, which may discourage or prohibit PSIamong young women [41]. Young women in the Northwestern, Southern, Northern and Eastern regions had higher odds of engaging in PSI than those in the Western region in Sierra Leone. Cultural attitudes towards sex, relationships, and gender roles can vary significantly across regions. Some regions might be more conservative, leading to less open communication about sex and sexuality, potentially affecting PSI rates [42]. Regional disparities in poverty, education levels, access to healthcare

and family planning services could all influence PSI rates. Lower socioeconomic status might be linked to higher rates of PSI due to limited access to contraception or education [43].

## Implications for Policy and practices

Since PSIamong young women in Sierra Leone is influenced by factors such as age, education, exposure to media, wealth index, region, and place of residence, there is a need for comprehensive sexual education and awareness programs that can address these issues and provide accurate and relevant information to young women. Such programs can also empower young women to make informed and responsible decisions about their sexual and reproductive health and rights. Family and community support can play a vital role in preventing or reducing PSIamong young women in Sierra Leone. Parents and guardians can provide guidance, counselling, and supervision to their daughters and encourage them to abstain from or delay sexual initiation until marriage. Community leaders and religious groups can also promote positive values and norms that discourage PSIand protect young women from sexual exploitation and abuse. Policymakers and practitioners should involve men and boys in the efforts to promote gender equality and prevent PSIamong young women. Men and boys should be educated about the benefits of delaying sexual debut, respecting women's rights, and choices, and sharing the responsibility for contraception and preventing sexually transmitted infections. Policymakers and practitioners should also focus on reducing poverty, banning sex with minors, banning non-consensual sex/forced marriage, increasing general education (not just comprehensive sex education) and employment opportunities for young women in Sierra Leone.

## Strengths and limitations

Data on sexual behaviour are often based on self-reporting, which can be influenced by social desirability bias and those biases are not likely to have changed over the three survey periods (2008, 2013, and 2019). Respondents may underreport or overreport certain behaviours due to societal norms or personal preferences, leading to potential inaccuracies. While multiple survey years provide a temporal dimension, the data are still essentially cross-sectional at each survey point. This limits the ability to establish causation and understand the dynamics of individual behaviour changes over time. While the survey aims to be representative, subgroups within the population may still be underrepresented or not captured adequately, limiting the generalizability of findings to specific demographic or geographic groups. Despite the limitations mentioned above, the study has specific strengths that need to be mentioned. First, using data from multiple survey years (2008, 2013, and 2019) allows for longitudinal analysis, providing insights into changes over time. This can help identify trends and patterns in PSI among young women in Sierra Leone. Also, understanding trends and associations can inform policies and interventions aimed at addressing challenges related to PSIamong young women in Sierra Leone. Policymakers can use this information to tailor programs to the population's needs.

## Conclusion

Our study has shown that PSI is prevalent among young women in Sierra Leone. Factors identified to be associated with PSI were age, education level, media exposure, wealth index, geographic region, and place of residence. Therefore, programs addressing PSI among young women should carefully consider these factors identified in the study. Also, it would be beneficial to implement more focused initiatives aimed at enhancing the financial empowerment of young women and providing them with comprehensive education on sexual and reproductive

health. This education should encompass information on the potential negative consequences of engaging in PSI such as unintended pregnancies, STIs, unsafe abortions and social stigma and discrimination, as well as promoting the adoption of abstinence and the utilisation of contraceptives.

## Acknowledgments

We are grateful to the MEASURE DHS Program for providing access to the dataset.

## Author Contributions

**Conceptualization:** Augustus Osborne, Castro Ayebeng, Camilla Bangura.

**Data curation:** Augustus Osborne, Castro Ayebeng.

**Formal analysis:** Augustus Osborne, Castro Ayebeng, Camilla Bangura.

**Investigation:** Augustus Osborne.

**Methodology:** Augustus Osborne, Peter Bai James, Richard Gyan Aboagye, Bright Opoku Ahinkorah.

**Project administration:** Augustus Osborne.

**Software:** Augustus Osborne.

**Supervision:** Augustus Osborne, Bright Opoku Ahinkorah.

**Validation:** Augustus Osborne, Peter Bai James, Richard Gyan Aboagye, Bright Opoku Ahinkorah.

**Visualization:** Augustus Osborne.

**Writing – original draft:** Augustus Osborne, Castro Ayebeng, Peter Bai James, Camilla Bangura, Richard Gyan Aboagye, Bright Opoku Ahinkorah.

**Writing – review & editing:** Augustus Osborne, Castro Ayebeng, Peter Bai James, Camilla Bangura, Richard Gyan Aboagye, Bright Opoku Ahinkorah.

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
