## [Decision Letter · Decision Letter 0]

4 Jul 2024

PONE-D-24-11274Trends, patterns, and associations of premarital sexual intercourse among never-married young women aged 15-24 in Sierra Leone.PLOS ONE

Dear Dr. Osborne,

Thank you for submitting your manuscript to PLOS ONE. After careful consideration, we feel that it has merit but does not fully meet PLOS ONE’s publication criteria as it currently stands. Therefore, we invite you to submit a revised version of the manuscript that addresses the points raised during the review process.  The comments from the reviewers, one of which was me, are below.  

We look forward to receiving your revised manuscript.

Kind regards,

Matt A Price

Academic Editor

PLOS ONE

Journal Requirements:

**Additional Editor Comments:**

Abstract, conclusion: I’m not sure how “Implementing collaborative endeavours to enhance young women's financial empowerment is imperative.” Follows from your results. Please clarify, or consider removing or modifying this statement

Lines 118+: You’re very explicit about how you create this variable, which is good – but please confirm that those who report sex, reported it before marriage (premarital, right?). Do you have date of marriage, or some other means to confirm they report sex prior to marriage? I imagine some report having had sex, but also after they got married

Line 168: you note that the prevalence of PSI increased “significantly”. How do you define “significantly”? This is typically accompanied by some statistical test result, but perhaps you could clarify.

Line 321: I don’t think “mitigate” is the right word here, I think you mean to say something along the lines of making PSI as safe and fulfilling as possible. Mitigate implies PSI is a problem, which I don’t think is your point.

Reviewers' comments:

Reviewer's Responses to Questions

**Comments to the Author**

1. Is the manuscript technically sound, and do the data support the conclusions?

Reviewer #1: Yes

2. Has the statistical analysis been performed appropriately and rigorously? 

Reviewer #1: Yes

3. Have the authors made all data underlying the findings in their manuscript fully available?

Reviewer #1: Yes

4. Is the manuscript presented in an intelligible fashion and written in standard English?

Reviewer #1: Yes

5. Review Comments to the Author

Reviewer #1: Overall, this paper is well written and well organized with appropriate statistical analyses. My main critique is that more should be said about why repeated cross-sectional analyses are important to examine PSI rates, from a conceptual perspective. Do the authors believe that there are some societal drivers that influence change over time in these rates? I am also curious to understand why rates differ by District. Are these differences constant over time? In the Discussion, the authors state that more research is needed to understand these differences, but can you provide a speculation? For example, might cultural differences explain some of this variation? Otherwise this is a solid study. More minor recommendations are as follows:

Line 100 - are there more recent population estimates for the country than 2015?

Line 103 - I believe "Creole" should be "Krio".

Lines 184-188 - What is the comparison group for media exposure?

Line 224 - In Table 2, the ordering of the model results is not what I expected. It would make more sense to include all covariates in Model 1 into Model 2, and then likewise into Model 3. This is my understanding of how hierarchical models are presented.

6. PLOS authors have the option to publish the peer review history of their article (what does this mean?). If published, this will include your full peer review and any attached files.

Reviewer #1: No

---

## [Author Response · Author response to Decision Letter 0]

16 Jul 2024

The Editor

PLOS ONE

15th July 2024 

Ref: PONE-D-24-11274

Title: Trends, patterns, and associations of premarital sexual intercourse among never-married young women aged 15-24 in Sierra Leone.

Response to Reviewers' comments 

Dear Sir/Madam, 

We want to express our sincere thanks for painstakingly reviewing our manuscript and providing valuable comments and suggestions. Please see our point-by-point response to the reviewers' comments and suggestions. Revisions are highlighted in blue in the revised manuscript.

RESPONSE TO REVIEWERS

Reviewer Comments:

Response: Thank you. We have done so. 

Response: Thank you. We have done so in the manuscript.

Response: Thank you. We have checked our references and are complete and there are no retracted references on the list.

Additional Editor Comments:

Abstract, conclusion: I’m not sure how “Implementing collaborative endeavours to enhance young women's financial empowerment is imperative.” Follows from your results. Please clarify, or consider removing or modifying this statement

Response: Thank you. We have removed it from our manuscript.

Lines 118+: You’re very explicit about how you create this variable, which is good – but please confirm that those who report sex, reported it before marriage (premarital, right?). Do you have date of marriage, or some other means to confirm they report sex prior to marriage? I imagine some report having had sex, but also after they got married

Response: Please, be informed that the analytical sample was restricted to never-married young women. Kindly refer to the topic.

Line 168: you note that the prevalence of PSI increased “significantly”. How do you define “significantly”? This is typically accompanied by some statistical test result, but perhaps you could clarify.

Response: Thank you. We have removed significantly from that sentence in our manuscript.

Line 321: I don’t think “mitigate” is the right word here, I think you mean to say something along the lines of making PSI as safe and fulfilling as possible. Mitigate implies PSI is a problem, which I don’t think is your point.

Response: Thank you. It now reads as therefore, programs to make PSI safe and fulfilling as possible among young women should carefully consider these factors.

Reviewer #1: Overall, this paper is well written and well organized with appropriate statistical analyses. My main critique is that more should be said about why repeated cross-sectional analyses are important to examine PSI rates, from a conceptual perspective. Do the authors believe that there are some societal drivers that influence change over time in these rates? 

Response: Thank you for your comment. By comparing data across multiple periods, we aimed to discern whether observed changes in PSI are consistent or transient, thus providing more robust insights into the dynamics of premarital sexual intercourse. Additionally, repeated cross-sectional studies help account for variations in demographic factors, ensuring that findings are representative and generalizable to the broader population.

I am also curious to understand why rates differ by District. Are these differences constant over time? In the Discussion, the authors state that more research is needed to understand these differences, but can you provide a speculation? For example, might cultural differences explain some of this variation? Otherwise this is a solid study. More minor recommendations are as follows:

Response: Thank you. We have now added that cultural attitudes towards sex, relationships, and gender roles can vary significantly across regions. Some regions might be more conservative, leading to less open communication about sex and sexuality, potentially affecting PSI rates[42]. Regional disparities in poverty, education levels, access to healthcare and family planning services could all influence PSI rates. Lower socioeconomic status might be linked to higher rates of PSI due to limited access to contraception or education[43].

Line 100 - are there more recent population estimates for the country than 2015?

Response: Thank you. We have provided a more recent population estimate in the manuscript. In 2024, Sierra Leone's population stands at 8,977,972, reflecting a 2.13% growth compared to the previous year[21].

Line 103 - I believe "Creole" should be "Krio".

Response: Thank you. Both is fine but we have changed it to Krio.

Lines 184-188 - What is the comparison group for media exposure?

Response: The comparison group is “those who were not exposed” as highlighted in line_187. Thank you.

Line 224 - In Table 2, the ordering of the model results is not what I expected. It would make more sense to include all covariates in Model 1 into Model 2, and then likewise into Model 3. This is my understanding of how hierarchical models are presented.

Response: We acknowledge your concern; however, this step-by-step approach allows for a comprehensive understanding of how different sets of variables (individual-level and contextual level factors) impact the outcome, leading to a robust and well-validated final model.

Kindly refer to a similar approach in other publications below:

https://journals.plos.org/plosone/article?id=10.1371/journal.pone.0297021

https://www.tandfonline.com/doi/abs/10.1080/23251042.2016.1197355

Etc. 

We hope that we have adequately addressed the reviewers' comments, and we look forward to receiving a favorable outcome on our paper. 

Yours Sincerely,

Augustus Osborne

Corresponding Author

---

## [Editor Report · Decision Letter 1]

8 Aug 2024

Trends, patterns, and associations of premarital sexual intercourse among never-married young women aged 15-24 in Sierra Leone

PONE-D-24-11274R1

Dear Dr. Osborne,

We’re pleased to inform you that your manuscript has been judged scientifically suitable for publication and will be formally accepted for publication once it meets all outstanding technical requirements.

Kind regards,

Matt A Price

Academic Editor

PLOS ONE

Additional Editor Comments (optional):

The reviewers' concerns have been addressed. Thank you.
---

## [Editor Report · Acceptance letter]

13 Aug 2024

PONE-D-24-11274R1 

PLOS ONE

Dear Dr. Osborne, 

I'm pleased to inform you that your manuscript has been deemed suitable for publication in PLOS ONE. Congratulations! Your manuscript is now being handed over to our production team.

Kind regards, 

on behalf of

Dr. Matt A Price 

Academic Editor

PLOS ONE